

# Timing, Causes, and Ecological Impacts of the 1991 Glacial Lake Outburst Flood at Rijieco in the Eastern Himalayas

Kaiheng Hu[1,2], Manish Raj Gouli[1,2,3], Hao Li[1,2,3], Yong Nie[1,2], Yifan Shu[1,2,3], Shuang Liu[1,2], Pu Li[1,2], Xiaopeng Zhang[1,2,3]

[1]State Key Laboratory of Mountain Hazards and Engineering Safety, Chinese Academy of Sciences, Chengdu, Sichuan 610041, China.
[2]Institute of Mountain Hazards and Environment, Chinese Academy of Sciences, Chengdu, Sichuan 610041, China.
[3]University of Chinese Academy of Sciences, Beijing 100049, China

*Correspondence to*: Kaiheng Hu (khhu@imde.ac.cn)

**Abstract.** We investigate and reconstruct a publicly lesser-known glacial lake outburst flood that happened at a moraine-dammed proglacial lake – Rijieco (27.963° N, 88.9° E) in the eastern Himalayas, Tibet, China, more than thirty years ago. Satellite images interpret that from 1977 to 1991, the surface area of Rijieco increased from 0.34 km$^2$ to 0.43 km$^2$, along with a reduction of 0.27 km$^2$ in the glacial area, and subsequently dropped to 0.24 km$^2$ after the GLOF. Further inspection of the

1991 images narrows down the date of occurrence of the Rijieco GLOF to between 21 September – 7 October 1991. The most probable triggering mechanism of the GLOF is an avalanche from the south-west part of the glacial lake because the local hydrometeorological data show precipitation of 100mm higher than the multi-year mean in the preceding two months and an elevated temperature anomaly in the month of the disaster, combined with a sufficient topographic potential of the slope. Based on the field-measured dimension of the dam gap (31 m high and 75 m wide) and local topography, it is estimated that nearly

6 million m$^3$ of impounded water was released during the GLOF. The reconstruction of the outburst flood with the HEC-RAS 2D hydrodynamic model shows that the flood peak discharge at the dam was 2900 m$^3$/s and then attenuated first to around 200 m$^3$/s at the alluvial fan and 45 m$^3$/s at the entrance to the Duoqing lake, an inland lake about 50 km away from the dam. The sudden release of the impounded water and large volumes of entrained sediment shortly expanded the Duoqing water area by 12.2% and strongly disturbed its margin vegetation. The transported sediments silted an area of 1.0×10$^6$ m$^2$ of the channel and

flood plain where new vegetation has not yet recovered. The harm of GLOFs to the lake ecosystem in the high-altitude Himalayan region may not be repaired in a short period. This study reveals short-term geomorphic impacts of GLOFs and noticeable but less mentioned long-term ecological impacts on a Tibetan inland lake system.

## 1 Introduction

The High Mountain Asia (HMA) - comprising both Himalaya and Hindu Kush- hosts the highest number of glaciers apart

from the polar region (Yao et al., 2012). These glaciers modulate the meltwater runoff of major river basins like Brahmaputra, Ganges, and Indus, on which billions of people depend for drinking, irrigation, and hydropower in concurrence with the



monsoon rainfall (Biemans et al., 2019; Immerzeel et al., 2013; Pritchard, 2019). In the wake of global warming, the climate of the Himalayas and Tibetan region has been impacted, resulting in a temperature increment of 0.42°C per decade - twice the global average - with heterogeneous precipitation trends (Yao et al., 2022). These changes have accelerated the rate of glacier

recession and led to negative cryosphere mass balance due to the loss of glacier mass (Nie et al., 2010, 2021; Wester et al., 2019). Such mass loss has contributed to the rapid increase in the number and size of glacial lakes globally (Haritashya et al., 2018; Nie et al., 2017; Shugar et al., 2020; Taylor et al., 2023). When water impounded in these lakes is suddenly released due to different triggering mechanisms (e.g., avalanche, rockfall, precipitation, and earthquake.), the massive amount of water and sediments are drained rapidly in the form of outburst floods, commonly called Glacial Lake Outburst Floods (GLOF) (Costa

& Schuster, 1987; Westoby et al., 2014a; Worni et al., 2014). GLOFs are among the worst natural hazards in high mountainous regions, potentially threatening 15 million people worldwide (Taylor et al., 2023). More than 600 GLOF events have been reported in the HMA, most of which occurred from moraine-dammed glacial lakes (Lützow et al., 2023; Shrestha et al., 2023). However, triggering mechanisms and processes for most reported events remain unconfirmed due to limited site-specific investigations and reconstructions of historical GLOF disasters (Zheng et al., 2021).

The Himalayan region is one of the most GLOF-susceptible areas (Taylor et al., 2023). As many as seventy-nine events have been recorded there, most of which were generated from the failure of moraine-dammed lakes (Lützow et al., 2023). The GLOF events are mainly concentrated in the Central and Eastern parts and are distributed across China, Bhutan, Nepal, and India (Nie et al., 2018). Some notable outburst floods in the Himalayas with substantial damages to the down-lying extents are from Cirenmaco (1964, 1981, and 1983), Luggye Tsho (1994), Chorabari (2013) and Gongbatongsha (2016) (Allen et al.,

2016; Himalaya & Watanabe, 1996; Sattar et al., 2022; Wang et al., 2018). The 1981 Cirenmaco event was the most devastating among others, claiming more than 200 lives and destroying bridges, roads, and hydropower, even towards Nepal's side (Chen et al., 2013; Xu, 1988). Likewise, Luggye Tsho GLOF in Eastern Himalaya, which occurred in 1994, killed 21 people and damaged Bhutan's houses, bridges, and agricultural lands. In fact, the Luggye Tsho outburst flood was found to activate GLOF from another immediately downstream lake and cause heavy damage (Himalaya & Watanabe, 1996). The latest transboundary

GLOF occurred in the China- Nepal border area from breaching a relatively small glacial lake called Gongbatongsha in 2016. The outburst flood cascaded through the Bhotekoshi River in Nepal, resulting in bank collapse and landslides. The flood severely damaged houses, roads, and infrastructures along the Bhotekoshi River, exceeding that year's seasonal monsoon flood effects (Cook et al., 2018). Another noticeable but rarely mentioned impact of GLOFs is their harm to the ecological environment by transporting vast amounts of water and sediment during a short period, covering the vegetation with sand and

rock particles, and ceasing their growth (Byers et al., 2019; Carpenter, 2017). Exceptional sediment inflow from GLOFs has long-term influences on vegetation, biological habitats, and water quality of downstream rivers and lakes.

In the Eastern Himalayas, the GLOF risk is comparatively higher than in other regions (Veh et al., 2020); therefore, correctly understanding GLOF drivers is essential for developing more robust hazard and risk mitigation strategies. Several GLOF hazards and risk assessment studies have been carried out in this region, considering present and future scenarios and possible

breach cases (Gouli et al., 2023; Hu et al., 2022; Rinzin et al., 2023; Sattar et al., 2023). Remote sensing tools have been widely



used in the Himalayas for monitoring glacial lakes through their spatiotemporal evolution (Haritashya et al., 2018; Nie et al., 2017; Wang et al., 2020) as well as to detect the historical GLOFs and their triggers (Nie et al., 2018; Veh et al., 2018). However, noises such as cloud cover in monsoons, ice, and shadow often impede detailed tracking of glacial lakes and, thus, GLOFs (Veh et al., 2018). Further, past GLOFs, such as those before 2000, are challenging to detect due to poorly quality

images (Nie et al., 2018; Zheng et al., 2021). To overcome such limitations and to reconstruct past GLOFs more realistically, interdisciplinary studies involving remote sensing, data from in situ instrumentation, and field investigations, followed by detailed hydrodynamic modeling and downstream impact assessment are essential (Byers et al., 2020; Nie et al., 2020; Peng et al., 2023). Correctly reconstructed GLOF events are vital to understanding the process chain and model calibration of future floods or watersheds that are identical in hydrology and geomorphology; therefore, leveraging them for accurate future hazard

assessments (Klimeš et al., 2014; Worni et al., 2014).

Most of the recent case studies of GLOFs demonstrate their short-term societal and geomorphic effects on downstream areas, such as valley incision, bank erosion, stream migration, and flood inundation (Duan et al., 2023; Miles et al., 2018; Zheng et al., 2021). Concerning the long-term impacts of GLOFs on the downstream landscape, we should choose those GLOF events that occurred decades ago to investigate. In this regard, the Rijieco glacial lake on the eastern Himalayas's north slopes served

as our study case. Our field evidence and historical satellite images confirm that a high-magnitude GLOF occurred from Rijieco glacial lake in 1991. This study aims to: a) carry out a retrospective analysis of Rijieco glacial lake and the glacier's evolution in the catchment between 1977 and 2020, coupled with the influence of climatic parameters and more focused on the 1991 GLOF event. b) simulate the outburst flood from the breaching of Rijieco glacial lake in the HEC-RAS 2D model using breach dimensions collected from the field survey and its validation using historical satellite imagery. c) demonstrate long-lasting

ecological impacts of the outburst floods on the river, downstream areas, and plateau's geomorphology. Our study reveals that the plateau ecosystem is vulnerable to GLOFs. After the glacial lake outburst disaster, vegetation and wetlands have not recovered in over 30 years. The harm to the lake ecosystem in the high-altitude Himalayan region is irreversible in a short period. In broader terms, this study aims to evaluate the long-term effects of the Rijieco GLOF event and reflect its understanding of future hazards and risk assessment of outburst floods.

**2 Study area**

The study area is located in Yadong County, the Tibetan Autonomous Region, China, at the junction of the Central and Eastern Himalayas (**Fig. 1**). The Tethyan Himalayan Sequence, including Paleozoic and Mesozoic carbonate and clastic sedimentary strata, dominate the regional lithology. The Kazu catchment containing the Rijieco glacial lake (27°57'56"N, 88°53'42"E) is an upstream branch of the Nian Chu River, flowing eastward into a bigger inland lake called Duoqing Lake in the southern

Tibetan Plateau (**Fig. 1a**). The whole catchment lies in China, with an area of 21 km$^2$, and the upper portion extends up to the China–India border. The topmost elevation of the lake's catchment is 7125 m.a.s.l, and Rijieco seats at 5400 m, with 0.25 m$^2$ area, as of 2020. There are three glaciers upstream of the Rijieco lake. The trunk glacier's terminus is about 600 m away from



the lake, as seen from the Bing satellite image (**Fig. 1b**). Outgoing water from the glacial lake flows through a narrow stream and mixes with other rivulets along the tributaries before finally debouching into Duoqing Lake. Before that, an alluvial fan

(commonly called Kazu alluvial fan) is located (**Fig. 1c**), where all the upstream flow spreads out. The fan is situated approximately 30 km away from the Rijieco Lake. Large-volume sediment sourced from the Kazu compress the wetland area of Duoqing.

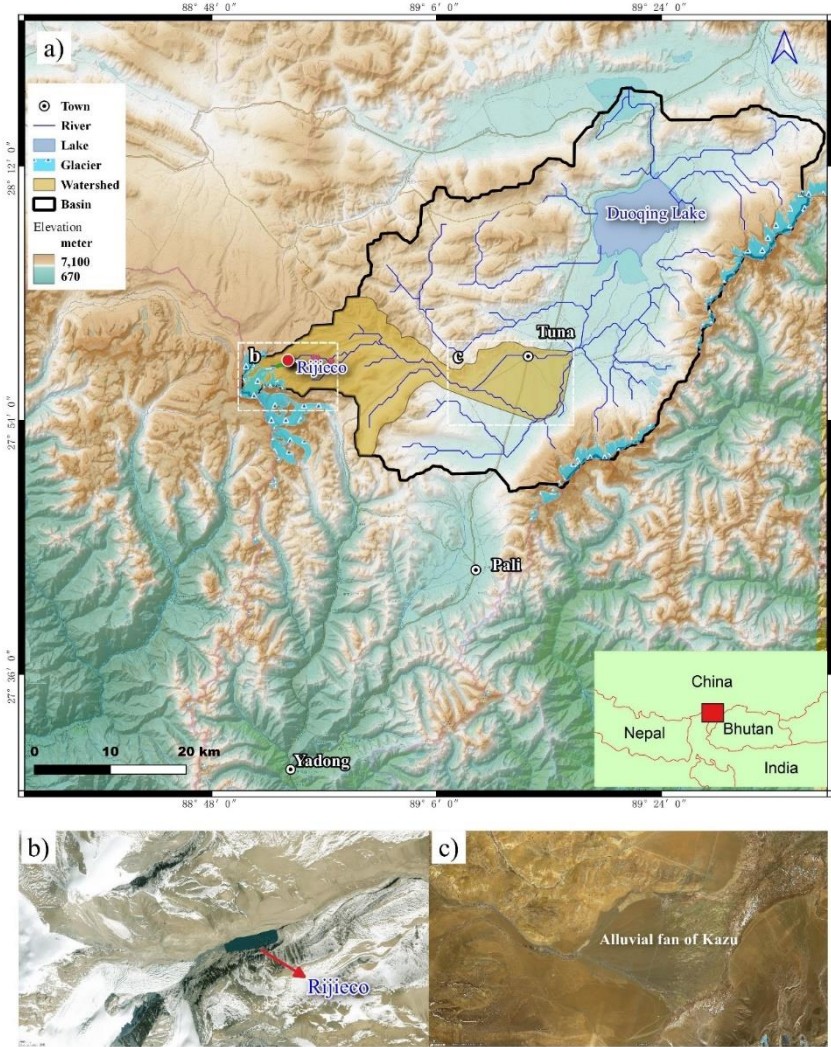

**Figure 1: Location map and satellite images of the study area: a) Rijieco glacial lake, Kazu watershed, and Duoqing lake in the study**
**region. b) The satellite image of the Rijieco Glacial Lake was sourced from the Bing Maps Aerial (the base image © 2024 Microsoft).**
**c) The Kazu alluvial fan satellite image is located 28 km downstream of the glacial lake and is sourced from SIWEIearth. The bottom**
**right inset shows the location of the study area.**

Westerlies and tropical monsoons dominate the climatic conditions in the study area. The rainy season is typically six months long, from early April to October. The Bengal typhoon, active in September and October, brings intense rainfall along the



Himalayan southern slopes (Anders et al., 2006; Bookhagen and Burbank., 2006). The annual rainfall to the south of the study area ranges from 937 to 2000 mm. But humid air currents become drier over the high Himalayan mountains. As per meteorological data from nearby Pali station (located at 4300 m a.s.l), this area's average yearly temperature and precipitation between 1957 and 2017 were 0.3 °C and 430 mm (**Fig. 2**). The maximum and minimum temperatures in the region were recorded as 19.3 and -30 °C in 1970 and 1965, respectively. Similarly, the area received a maximum daily precipitation of 130

mm in 2009. Overall, the temperature and precipitation are increasing, although the precipitation shows a slightly decreasing trend after the 1980s.

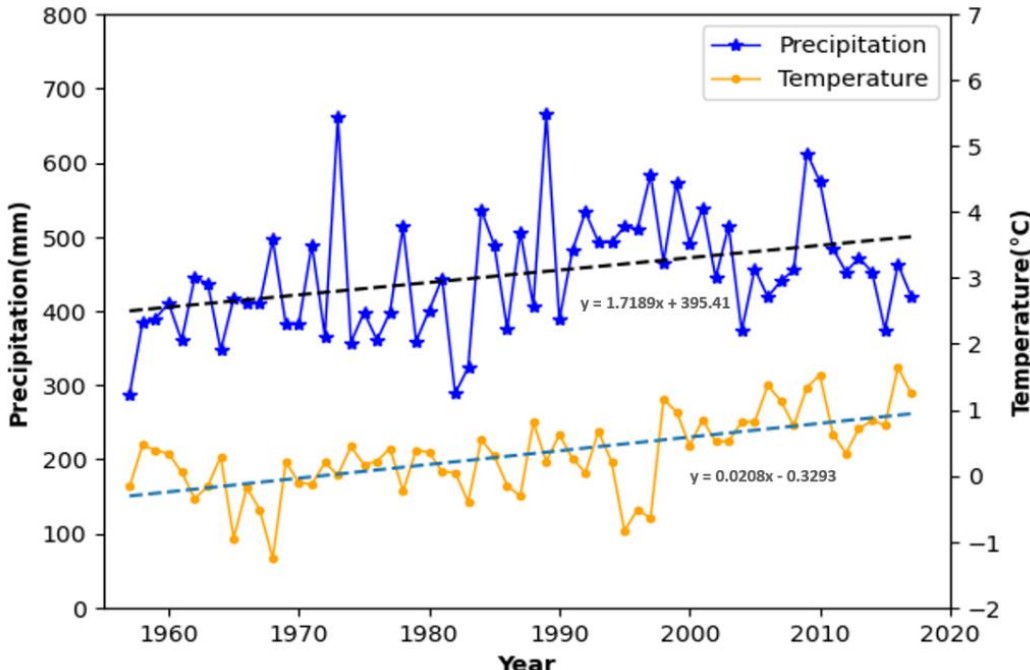

**Figure 2: Interannual temperature and precipitation variation in the study region between 1957 and 2017. These climatic records were taken from the Pali in-situ station, the nearest meteorological station close to Rijieco glacial lake.**

**3 Materials and Methods**

**3.1 Data sources**

An integrated methodology consisting of field survey, Remote Sensing (RS), Geographic Information System (GIS), and two-dimensional (2D) dam breach simulation is applied to figure out the evolution process of the glacial lake, the potential mechanism of the lake outburst, and the characteristics of the outburst flood. Our study mainly uses three kinds of data: remote-

sensing images, DEMs, and meteorological data (**Table 1**). Historical satellite images and digital elevation data from open data sources such as USGS Earth Explorer, Bing Maps, and Google Maps are used to extract the Rijieco area, the dam size, GLOF's inundation, etc. The field survey provides first-hand information on the GLOF deposition, the dam type, and the



dimensions of the dam gap. Detailed UAV images of the lake and the channel were captured in the field survey. The geomorphic conditions around the Rijieco detected from the satellite remote-sensing images are reliable when combined with interpreting the field UAV images. The DEMs used in dam-break modeling, calculations, and outburst flood simulation are High Mountain Asia (HMA) 8 m DEM (Shean, 2017) and NASADEM with 30 m resolution.

**Table 1: Different datasets, their coverage, and sources used in this study.**

| Data Type | Agency | Coverage/ Released Date | Sources |
|---|---|---|---|
| Satellite images | Landsat (Missions 2,5,7 and 8) | 1977-2020 | https://earthexplorer.usgs.gov/ |
| UAV images | DJI Mavic 2 | June, 2023 | |
| Meteorological Data | ERA5 hourly data | 1985-2016 | https://cds.climate.copernicus.eu/cdsapp#!/dataset/reanalysis-era5-land?tab=overview |
| | In situ Station | 1957-2017 | China Meteorological Administration (CMA) |
| HMA 8 m DEM | NSIDC - NASA | 2017 | https://nsidc.org/data/highmountainasia |
| 30 m NASADEM | NASA | 2019 | https://www.earthdata.nasa.gov/esds/competitive-programs/measures/nasadem |

Abbreviations: ERA, European Environment Agency; HMA, High Mountain Asia; DEM, Digital Elevation Model; NSIDC, National Snow and Ice Data Center (NSIDC); NASA, National Aeronautics and Space Administration

## 3.2 Interpretation of the satellite images

To investigate the evolution of Rijieco Glacial Lake and the contributing glaciers, seven cloud and seasonal snow-free Landsat (2,5,7, and 8) images were acquired from 1977 to 2020 (Refer Table 1). Landsat data were used for analysis since they have the most extended temporal coverage spanning more than forty years (Roy et al., 2014) and high-resolution images during our analysis period. The images were obtained during the post-monsoon season (between October and December) and were used to outline the extent of the glacial lake and glaciers. The images were generally processed for a decade; however, since the GLOF date is uncertain, additional images were collected for detailed analysis. The glacial lake was carefully identified and delineated manually by digitizing the false color composite images. Likewise, glaciers were semi-automatically outlined using the Normalized Difference Snow Index (NDSI), which is a ratio of difference and summation between the green and short-wave infrared bands (https://www.usgs.gov/landsat-missions/normalized-difference-snow-index). Then, we selected an optimal threshold to segregate glaciers from other surfaces. Later, manual post-correction was applied for cloud and mountain shadows to improve the glacier delineation using false color composite images, which a single expert undertook. Debris-covered glaciers were manually demarcated using Google Earth, and the glacier extent was refined with the help of Randolph



Glacier Inventory (RGI) V 6.0 (Pfeffer et al., 2014). The error in glacier extent was calculated as the product per pixel and the glacier polygon's perimeter (an average area of one-pixel buffer inside and outside) (Racoviteanu et al., 2015). Similarly, the glacial lake's uncertainty was obtained as the product of a half-image pixel and the glacial lake polygon's perimeter, as Fujita et al. (2009) suggested.

To clarify the influence of the GLOF on the Duoqing lake, the water bodies of the Duoqing are extracted from the 1988-1994 Landsat5 images by using four indexes of remote sensing including NDVI (Normalized Difference Vegetation Index), EVI (Enhanced Vegetation Index), NDWI (Normalized Difference Water Index), and mNDWI (modified NDWI) defined as follows (Lu et al., 2023):

$$NDVI = \frac{\rho_{NIR} - \rho_{Red}}{\rho_{NIR} + \rho_{Red}} \tag{1}$$

$$EVI = 2.5 \times \frac{\rho_{NIR} - \rho_{Red}}{1 + \rho_{NIR} + 6 \times \rho_{Red} - 7.5 \times \rho_{Blue}} \tag{2}$$

$$NDWI = \frac{\rho_{Green} - \rho_{NIR}}{\rho_{Green} + \rho_{NIR}} \tag{3}$$

$$mNDWI = \frac{\rho_{Green} - \rho_{SWIR}}{\rho_{Green} + \rho_{SWIR}} \tag{4}$$

where $\rho_{Blue}$, $\rho_{Green}$, $\rho_{Red}$, $\rho_{NIR}$, and $\rho_{SWIR}$ are the surface reflectance values of different bands in Landsat remote sensing data. We distinguish two kinds of pixels: water and non-water pixels. A pixel is classified as a water pixel if it meets the following
criteria: NDWI > -0.1, mNDWI > 0.1, EVI < 0.1, mNDWI > NDVI or mNDWI > EVI. Otherwise, the pixels not meeting these criteria are classified as non-water pixels. Changes in the lake area and shoreline can be known when the water pixels of Duoqing Lake are identified in the Landsat images.

**3.3 Hydrometeorological data analysis**

To investigate hydrometeorological conditions before the Rijieco GLOF and compare them with long-term climatic data
(1957-2017), we used daily precipitation and temperature data recorded by an in-situ station in Pali – located at an elevation of 4300 m.a.s.l. The station is 35 km from the Rijieco glacial lake and is the only nearby station to be used for analysis in this study. Additionally, we used ECMWF Climate Reanalysis (ERA 5) - Land Hourly data (https://cds.climate.copernicus.eu/cdsapp#!/dataset/reanalysis-era5-land?tab=overview) to evaluate the precipitation and temperature data for the lake catchment area from several days before our hypothesized outburst date. The resolution of the
ERA-5 satellite product is 0.1 arc seconds, and it provides hourly data; therefore, it was found to be more comprehensive than other satellite data. Moreover, due to the considerable spatial heterogeneity of precipitation in the Hindu Kush Himalaya region (Sabin et al., 2020), satellite information was used to check the relevance of station data. Several time series variations of collected climatic parameters were scrutinized on a periodical basis (annual, monthly, and daily) to interpret the role of hydrometeorological circumstances in the short and longer term and its role on the local cryosphere evolution and the outburst
event.



### 3.4 Field observations

We carried out one-day fieldwork from the middle stream of Kazu to the Rijieco's dam on June 27, 2023 (**Fig. 3a**). Portable measuring equipment such as handheld GPSs, a laser rangefinder, and two DJI Mavic 2 drones were used to measure local topography and cross sections, position some interesting points (the locations that retain evidence of the GLOF), and obtain aerial photos. During our field visit, we viewed a gap of 75 m in top width and 31 m in height on the left side of the terminal moraine dam, indicating at least one lake outburst flood. The lake's terminal dam is about 80 m high, 320 m long, and 50 m wide (**Fig. 3b and 3c**). The dam is made up of moraines with wide size ranges from sand to boulders. Part of the flood sediment with an area of 85000 $m^2$ deposits immediately behind the dam. The surface of the deposits is about 9 m above the present channel bottom (5400 m a.s.l). Melting water from the upstream glaciers inputs into the lake and drains through the gap at a flow discharge of 2-3 $m^3$/s.

The lake is oriented towards the south-west direction. The feeding glacier is on the western side, almost perpendicular to the lake. The lake disconnects with the tongue of the glacier, which is covered with only a small amount of moraine and fully distributed with fissures. Fine sediments are deposited between the glacier and the lake (**Fig. 3c**). The ice/snow accumulated areas are located far from the water body, with a low probability that an avalanche triggered from it would directly enter the lake. Steep lateral moraines are located on the lake's northern and southern sides, with a mean slope of 20 ° in the inner part. At least three periods of later moraines can be identified on both sides of the lake, corresponding to three recent glacier advances. A straight shoreline below the lowest later moraine is considered the maximum water level before the lake outburst. The vertical height is 55 m between the shoreline and the second lateral moraine (**Fig. 3b**). According to this previous water level, the lake's largest area was approximately 0.46 $km^2$. No large-scale landslides or rock avalanches were observed around the lake. The lake's sediment sources are till, outwash, colluviums, and rockfall.

Flood sediments spread over the whole channel from the dam to the middle stream (**Fig. 3a**), implying the outburst flood probably transformed into debris flows. The total depositional area in the survey reach is about 1.0×106 $m^2$. Although the flood or debris-flow deposits look fresh, it is observed that the deposits were formed at least before 1996 when compared with the Landsat satellite images in 1996. Furthermore, to study vegetation dynamics before and after the outburst event, the NDVI index was calculated between two cloud-free images of 1989 (Pre-GLOF) and 1991(Post-GLOF). We found that the NDVI value in the floodplain before the Rijieco GLOF was about 0.1, indicating the presence of sparse vegetation. However, after the GLOF, the NDVI value decreased to -0.07, representing the coverage of the area by sediments and bare soil. Many slope failures on both channel sides are developed (**Fig. 3d and 3e**). We found remnants of the deposits with the surface almost parallel to the channel 4.2 km downstream of the dam (**Fig. 3e**). The cross-section is about 130 m wide, and the deposition is about 3.0 m high. We estimated that the average velocity in this section was 4.0 m/s using a Manning-type empirical formula (roughness = 0.15, channel slope = 0.083, flow depth = 3.0 m). If the flow section was rectangular, the peak discharge was about 1560 $m^3$/s. The GLOF's sediment covers the floodplain meadows in the middle stream (**Fig. 3f**).



**Figure 3: Field photographs were taken during the survey on June 27, 2023. a) the locations of Figure 3 b-c, the red arrow indicates the camera's orientation (the base image © 2024 Microsoft). b) UAV image of the terminal dam viewed from the glacial lake. c) A UAV image of the dam's breach viewed from downstream shows the glacial lake, parent glaciers, and surrounding massifs with snow. d) Photo of bank failures and flood deposits 3.7 km downstream of the dam. e) Photo of bank failures and flood deposits 4.2 km downstream of the dam. f) The sediment deposits spread over the floodplains resulted from the GLOF.**

## 3.5 Reconstruction of Outburst Flood

The reconstruction of the outburst flood was carried out using an open-source hydrodynamic tool called HEC-RAS V 6.3.1. (refer https://www.hec.usace.army.mil/software/hec-ras/). The HEC-RAS tool has already been used to successfully reconstruct several GLOF events in the Chinese Himalayas (e.g., (Peng et al., 2023; Sattar et al., 2022). Here, we used a 2D



overtopping dam break module in HEC-RAS due to its suitability for alpine valleys with steep gradients (Pilotti et al., 2020).
The 2D unsteady flow model is based on sub-grid bathymetry and uses an implicit finite volume scheme to solve the Shallow
Water Equations (SWE) (Brunner, 2020).

This study mainly used a 30-m resolution Digital Elevation Model (DEM) named NASADEM as a terrain model. It is a state-
of-the-art global DEM produced after improving the Shuttle Radar Topography Mission (SRTM) DEM on several aspects like
elevation control, filling of voids, and addition of data that was unavailable before
(https://www.earthdata.nasa.gov/esds/competitive-programs/measures/nasadem). We also used the High Mountain Asia
(HMA)- a free 8 m high-resolution DEM for glacierized regions- for the initial dam break modeling and detailed overview of
the terrain (Shean, 2017). Despite the availability of such high-resolution DEM for the area, we could not consider it for
downstream hydrodynamic modeling due to large data voids. Therefore, NASADEM was used to model the downstream flood
propagation.

We modeled the glacial lake as a storage area and the flow area as a mesh of 30 m each. The moraine dam was represented as
an SA/ 2D tool that connected the glacial lake with flow mesh. Similarly, the computational time step was taken at 3 seconds.
In this model, we mainly used Manning's roughness value of 0.15, which was based on surface analysis (Chow, 1959) and past
studies (Maskey et al., 2020; Wang et al., 2018). Besides that, we also carried out a sensitivity analysis with other values, 0.07
and 0.1. Due to a lack of detailed bathymetry data, the volume was computed using the latest empirical relation given by Zhang
et al. (2023). The relation was derived from the bathymetric data of 59 proglacial lakes in the Himalayas. The dam breach
parameters like breach formation time (Tf), bottom width (Bw), and side slopes (H: V) were calculated using the empirical
relationship (Eq. (5) and (6)) developed by Forehlich (1995) as illustrated below:

$$B_w \ = \ 0.1803 \ * \ K_o \ (V_W)^{\ 0.32} \ * \ h_b^{\ 0.19} \tag{5}$$

$$T_f \ = \ 0.00254 \ (V_W)^{\ 0.53} \ * \ h_b^{\ -0.9} \tag{6}$$

Where $K_o$ = 1.4 for overtopping, $V_w$ = volume of water stored in the lake, and $h_b$ = Breach height
The breach height was measured at 31 m during the field visit. Using the Froehlich's equation (Froehlich, 1995), the top breach
was computed as 71 m, almost similar to the value (74 m) measured during the field visit. The range of breach parameters also
fits with the measurements from the HMA DEM, which gave the top width and breach height at 73 m and 30 m, respectively.
Since all the values lie within the plausible range of ± 3 m, using Froehlich's equation was justified for our study. The
downstream routing of the outburst flood was carried out up to 38 kilometers until it reached the runout zone and then
debouched into the inland lake.



# 4 Results

## 4.1 Evolution of the glacial lake and glacier

We analyzed Landsat imagery from 1977 to 2020 to perceive the variation of the glacial lake and glaciers (**Fig. 4**). The Rijieco Lake, connected to the trunk glacier, and the lake area increased before 1990. After 1991, the lake shrunk and disconnected
from the trunk glacier in 2020 (**Fig. 4f**). Since 1977, the upstream glaciers have been retreating significantly. In the 2010 image, the tributary glaciers on both sides of the lake disappeared. The melting water from the disappeared glaciers did not expand the lake, implying that the dam breached before 2010.

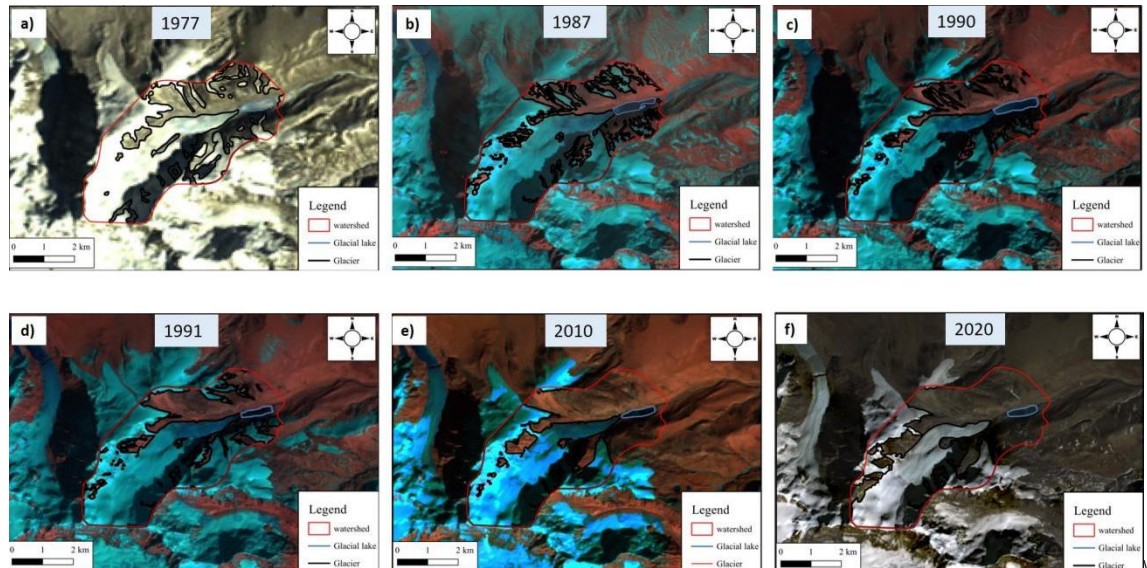

**Figure 4: Glacial lake and glaciers delineated from Landsat satellite imagery a) in 1977, b) in 1987, c) in 1990, d) in 1991, e) in 2010,**
**f) in 2020. The labels on the top of the image indicate the year from which the images were extracted.**

Downstream Landsat satellite images in 1991 show clear evidence of the outburst flood of the glacial lake (**Figure 5**). On September 21, 1991, the image shows widespread green lands on the alluvial fan and two channels on the north and south edges of the fan (**Fig. 5a**). Evidently, traces of fresh flood water emerged in the Landsat image of the Kazu alluvial fan on October 7, 1991 (**Fig. 5b**) and disappeared in the later images (**Fig. 5c**). This means that the Rijieco glacial lake outburst
occurred between September 21 and October 7, 1991. However, we could not verify the exact date due to a lack of day-wise satellite images and no information among the residents. Based on the clear traces of floodwater, we propose that the flood occurred at the end of 1991-October's first week~possibly October 7. The gray areas in **Fig. 5c** indicate that the outburst flood deposited large amounts of sediment on the fan apex, leading to the north channel disappearing and the south channel migration from the south to the north. It is also observed from the 1991-10-23 image that the Duoqing lake wetland increased, and the
green lands on the apex and margin of the fan was inundated two or three weeks after the GLOF event.





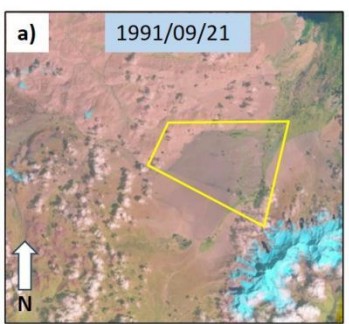 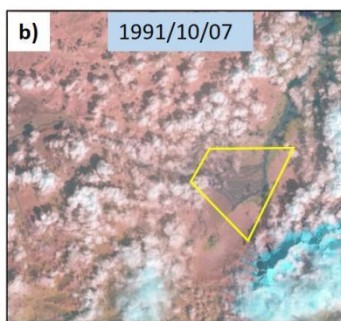 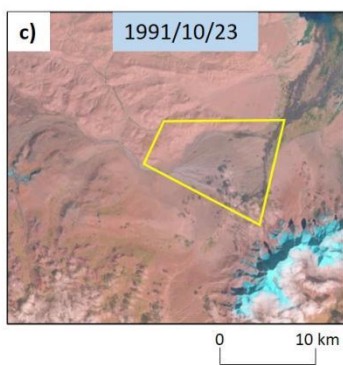

**Figure 5: Landsat satellite images show a) The alluvial fan is located around 30 km downstream of the glacial lake with no flood traces. b) Clear traces of fresh flood water can be observed in the October 7, 1991 image. c) Dry debris can be seen on the alluvial fan on October 23's cloud-free images.**

An advanced object-based mapping method (Nie et al., 2017; Nie et al., 2020) is used to estimate the Rijieco's surface area. The initial surface area of the lake was $0.34\pm0.041$ km$^2$ in 1977. Then, the lake started to expand, and just before the 1991 GLOF, the area increased by 26% ($0.43\pm0.048$ km$^2$); the maximum extent the lake ever reached was close to the field measurement (**Fig. 6**). During the GLOF event, the lake partially breached, and the area was reduced to $0.242\pm0.035$ km$^2$ – a sharp reduction of 44%. On the other hand, the glacier coverage was $15.46\pm0.89$ km$^2$ in the beginning, and by 1991, it had

reduced to $15.195\pm0.865$ km$^2$. After 1992, the lake area expanded to $0.275\pm0.041$ km$^2$ by 2000, and then it started to reduce at the 0.002 km$^2$/decade rate. Finally, by 2020, the glacial lake attained an area of $0.244\pm0.035$ km$^2$. In contrast, the glacier's reduction rate was as before (0.02 km$^2$/year) during the outburst, but it increased rapidly to 0.32 km$^2$/year after 1991's GLOF up to 2000. Since 2000, the glacier retreat rate was 0.016 km$^2$/decade up to 2010, and the proportion slightly increased to 0.11 km$^2$/decade after 2010. As of 2020, the final glacier area was recorded at $11.374\pm0.65$ km$^2$. Regarding the glacier termini

connection with the lake, the glacier's terminus was connected to the glacial lake during the 1991 GLOF and disconnected afterward.



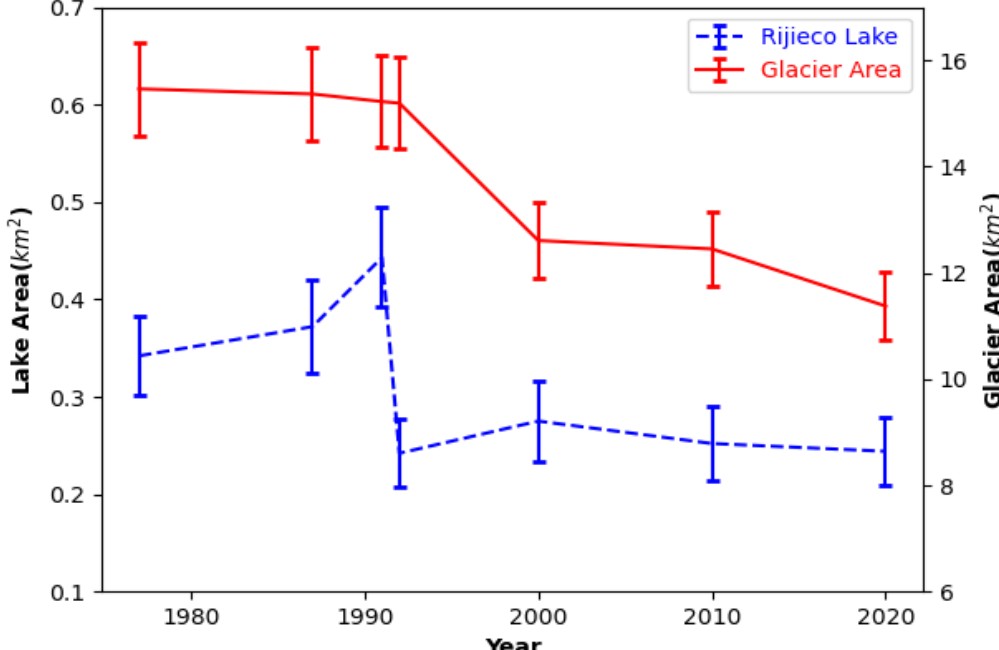

**Figure 6: Evolution of Rijieco Glacial Lake and its feeding glaciers from 1977 to 2020. The area uncertainties/errors are computed by multiplying the perimeter of glaciers/glacial lakes with half of the cell resolution.**

The glacier recession phenomenon shows that the glacier mainly retreats towards the terminus. We carried out a correlation test (Pearson's linear correlation coefficient) between the glacier area and the lake surface area over forty years, yet we did not find a strong correlation between the two areas. But when we evaluated the correlation before the GLOF, we observed a strong negative correlation of 0.99, indicating that the shrinking glaciers contributed additional meltwater into the glacial lake. Meanwhile, after the GLOF, there is a poor correlation, which we assume is due to the continuous flow of water from the

breach, even when the glacier melts, which supplies a substantial amount of water to the glacial lake.

**4.2 Potential Triggering Mechanism behind the GLOF**

From the detailed analysis of the lake and glacier's evolution, field investigation, and interpretation of historical imagery, we substantiate three likely causes behind the occurrence of Rijieco GLOF. The possible scenarios are presented chronologically based on their maximum likelihood (**Fig. 7**). Scenario A) potential avalanche from the slopes adjacent to the glacial lake (**Fig.**

**8**), Scenario B) Overtopping of the moraine dam due to the displacement waves triggered by glacier calving, and Scenario C) Failure of the moraine dam due to piping and expansion of internal drainage conduits. We propose 'Scenario A' as the most likely outburst trigger because several massifs surround the glacial lake and are covered with huge chunks of ice (**Fig. 8a**). The field survey discloses several slopes with huge crevasses, hanging ice, and fresh-colored glaciers (**Fig. 8b, 8c, and 8d**), signifying the ice avalanches as a regular process in the Rijieco basin. Among the neighboring slopes, we postulate the one

located immediately towards the south-western part (**Fig. 8b and its inset**) was liable for ice avalanche into the Rijieco lake





since it has a substantial area and satisfies the topographic potential criteria (located at slope >45° and <60° and reach angle >14°) as suggested by Rounce et al. (2017) and Allen et al. (2019). As the lake had reached its historical maximum surface extent at that time (**Fig. 4d**), it was already holding water up to the rim; thus, even a minimal wave height would have overtopped the moraine dam without sufficient freeboard. Apart from the ice mass, we also observed cracks and fractures on the permafrost

slopes (**Fig. 8e, 8f, and 8g**), and a fresh scarp and rockfall evidence were witnessed on the rock mass that was located just above the lake terminus (**Fig. 8e**), but mass inflow such scale does not seem sufficient to trigger GLOF from the glacial lake. Likewise, 'Scenario B' is another probable cause behind the breaching of the dam. **Figure 4b** shows that the glacier termini were well connected with the upstream end of the glacial lake when the GLOF took place. Though there is no high-resolution imagery for that period through which glacier crevasses can be visualized, it is reasonable to hypothesize that both contact of

the lake with steep glaciers and thermal undercutting of subaqueous ice due to increased water temperature in the summer period (Sakai et al., 2009) are possible bases of calving. Finally, we propose 'Scenario C' seepage/piping as another failure mechanism since this internal erosion mechanism is responsible for the failure of moraine dams due to increased hydraulic gradient (Neupane et al., 2019). In the Landsat satellite image in February 1991 (**Fig. 9a**), we spotted a small pool-like structure below the lake's spillway before the GLOF; however, it disappeared in the images acquired afterward (**Fig. 9b**). It is rational

to assume that the pond formed due to the water seeping through the dam, thus holding piping as one of the liable factors for the outburst. We ruled out the earthquake criteria because there were no significant earthquakes in the region or its surroundings when the flood occurred.

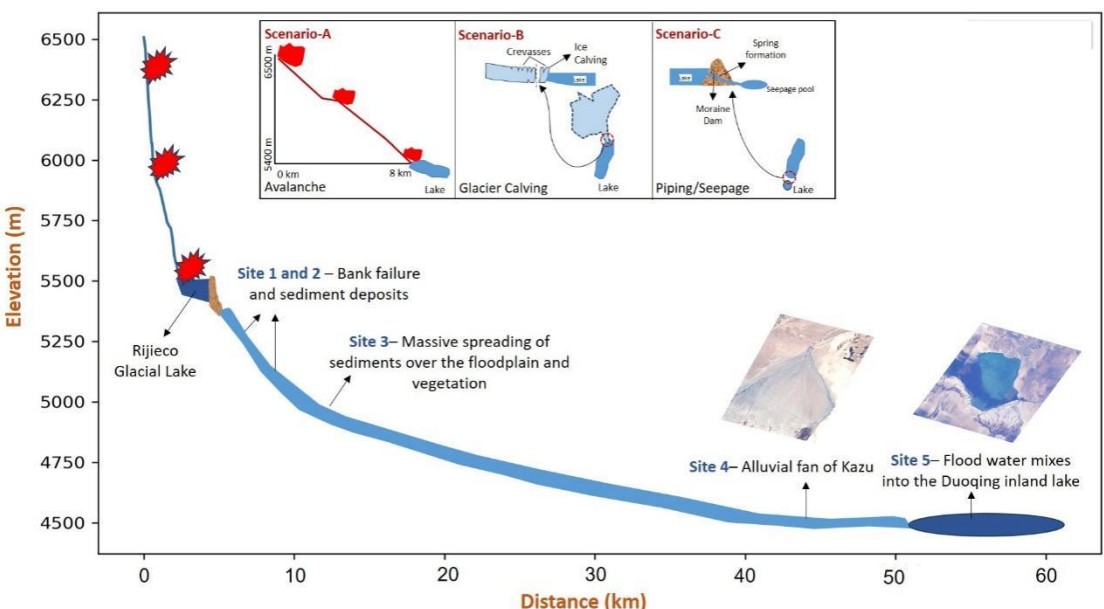

Figure 7: **Three possible scenarios behind the occurrence of Rijieco GLOF.**






**Figure 8: Field assessment photographs taken on June 27, 2023 show the Rijieco lake's surrounding features that make it susceptible to GLOFs. a) The total view of the Rijieco watershed and the susceptible massifs. b) A potential avalanche site with an ice crack from which is anticipated to have triggered the 1991 Rijeico GLOF (the highlighted zone in the inset). c) An adjacent slope with massive ice cracks. d) A massif towards the northwest side of the glacial lake with hanging ice. e, f, and g) Permafrost slopes and**
**rockfall potential zones with existing cracks on the rock mass.**



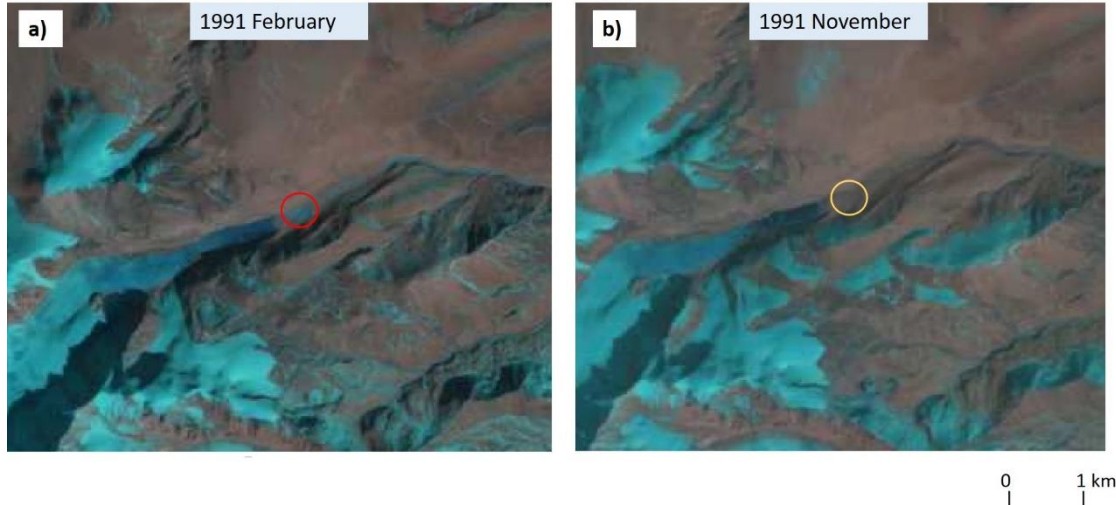

**Figure 9: Landsat images in February and November 1991 a) before the GLOF show a blue spot indicating a probable seepage location highlighted by a red circle b) After the GLOF, the blue spot disappears (inside yellow circle).**

### 4.3 Flood Reconstruction

During the 1991 GLOF, a total volume of 6 million cubic meters of water was estimated to have been released during the partial breaching of the lake. From the Froehlich (1995) relationship, the breach formation time was obtained at 30 minutes for given breach dimensions (W- 75, H-31), and upon simulation, the peak discharge at the dam site was calculated at 2900 $m^3$/s (**Fig. 10a**) and 40 $m^3$/s (**Fig. 10b**) at the outlet. Our results illustrate that it took about four hours to drain the given volume of water entirely from the glacial lake with the sharp rising limb of flood hydrograph and a little flat recession limb (**Fig. 10a**).

The inundation results show that the flood attained maximum depth and velocity of 10.8 m and 8.4 m/s across the modeling range. The flood gained maximum depth at 10 km downstream from the glacial lake – the area characterized by sudden contraction after a broad floodplain (**Fig. 11**). Just below the moraine dam of the Rijieco Lake, for a few distances, the gradient is very steep; therefore, the maximum velocity is observed 2 km below the glacial lake. When we routed the flood towards the downstream part, the inundation depth and discharge were impeded due to the receding gradient and broadening river width.

The mean flood depth and velocity in the mid-stretch of the stream were 5-8 m and 1-3 m/s, with a lateral inundation extent of 100 m. As the flood rushed towards the apex of the alluvial fan positioned at about 28 km downstream of the glacial lake, the flow braided over the alluviums (**Fig. 11**). The surge took six hours to reach there. As it spread over the sediment deposits, the resulting discharge attenuated first to around 200 $m^3$/s at the apex and then 45 $m^3$/s (**Fig. 10b**) at the fan's apron before flowing into the Duoqing inland lake. The average flood depth and velocity over the alluvial fan are 0.5 m and less than 0.2 m/s. The

Landsat 5 satellite image of October 7, 1991 (**Fig. 5b and 5c**) also shows the dispersion of flood and water being channelized into the inland lake. From the simulation and reference satellite images, we determined that first, the alluvial fan and second, the large lake downstream aided in reducing the impact of GLOF on low-lying settlements.



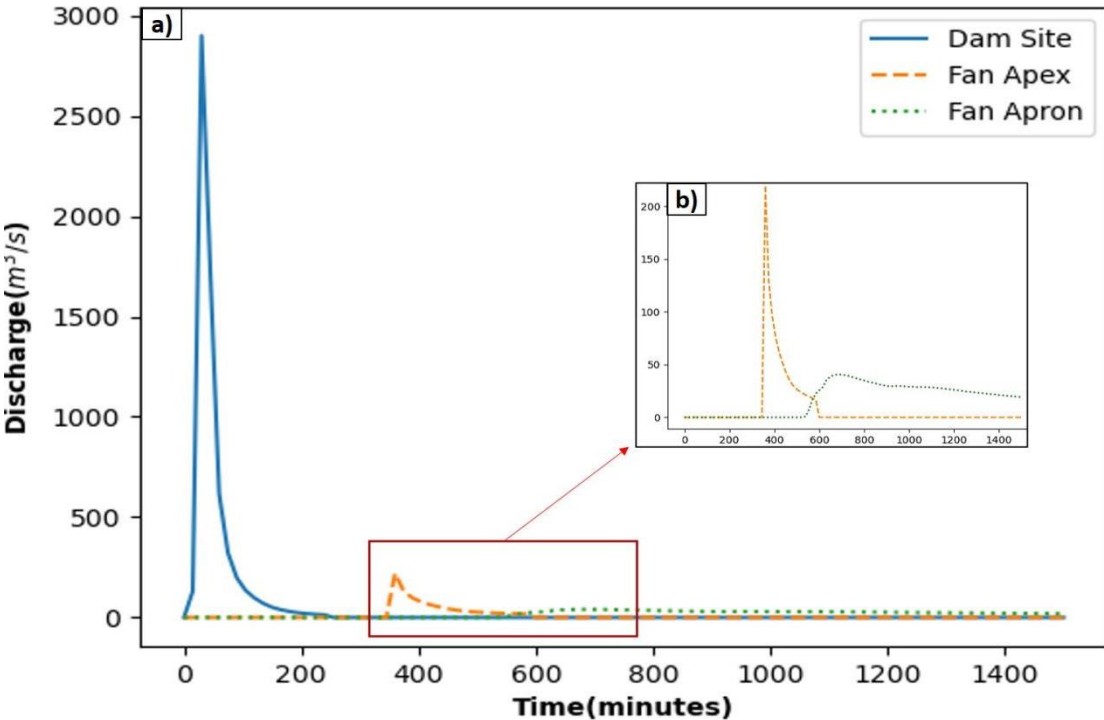

**Figure 10: Flood hydrograph of the reconstructed GLOF a) Discharge at the dam site with a peak flow of 2900 m³/s after thirty**
**minutes of the breach initiation. b) The inset shows an enlarged view of the flood hydrograph 30 km downstream of the dam site, where the peak flood reached after nearly six hours with a maximum discharge of greater than 200 m³/s and then to about 45 m³/s before terminating into an inland lake.**





**Figure 11: Flow depth and extent of the 1991 Rijieco GLOF were reconstructed using a two-dimensional dam break model (Upper Figure). The lower Figure shows the flow velocity and distribution of the GLOF. Manning's roughness (0.15) and bulking factor (1.3) were adopted for the simulation. The insets in the respective section show depth and velocity distribution in different reaches. (the base images © Esri, Maxar, Earthstar Geographics, and the GIS User Community)**





### 4.4 Downstream inundation and geomorphological impacts

Water and sediment transported downstream by the Rijieco GLOF greatly influenced the Duoqing lake. Large amounts of water and sediment poured into the Duoqing in a short time. Compared to the 1991/09/21 and the 1991/10/23 images, parts of the grassland and wetland were destroyed by the GLOF, especially at the edges of the alluvial fan and the entrance to the Duoqing (**Fig. 5**). The water area of the Duoqing increased, but the GLOF's sediment silted a small portion of the Duoqing. Most sediment comprises sand, granules, and pebbles (**Fig. 12**). We do not have information on the damages caused by the outburst of flood to the downstream community and infrastructures.

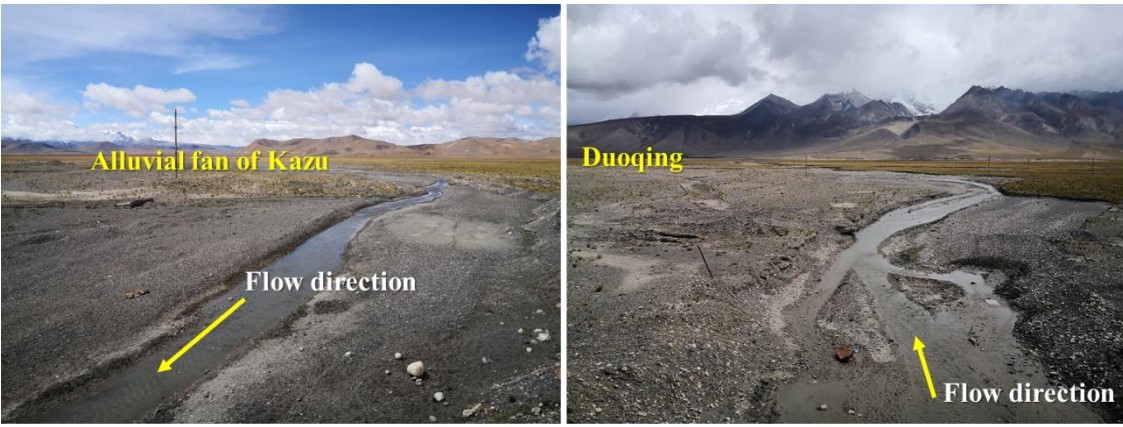


**Figure 12: Photos of the Kazu alluvial fan taken on September 17, 2020. a) sediment deposited on the alluvial fan, b) the stream flowing into the Duoqing lake.**

By using the four remote-sensing indexes (NDVI, EVI, NDWI, and mNDWI), a change in the area of Duoqing's water body was detected before and after the 1991 GLOF (**Fig. 13**). The water area gradually increased from 73 km$^2$ in 1988 to 78-79 km$^2$

in 1989 and 1990. In 1991, the lake significantly expanded to 87.5 km$^2$, about a sharp increase of 12.2% (**Fig. 13c**). The GLOF discharge caused the abnormal expansion. Subsequently, the water area was reduced to 72.6 km$^2$ in 1992 because the impounded water by the Rijieco was released, and a large amount of sediment accumulated upstream of the Duoqing in the GLOF event. After the GLOF (1992-1994), the south shoreline of the Duoqing moved further north than before the GLOF (1988-1990) (**Fig. 13b**), indicating a large amount of sediment was delivered from Kazu watershed and deposited in the

Duoqing lake by the GLOF.





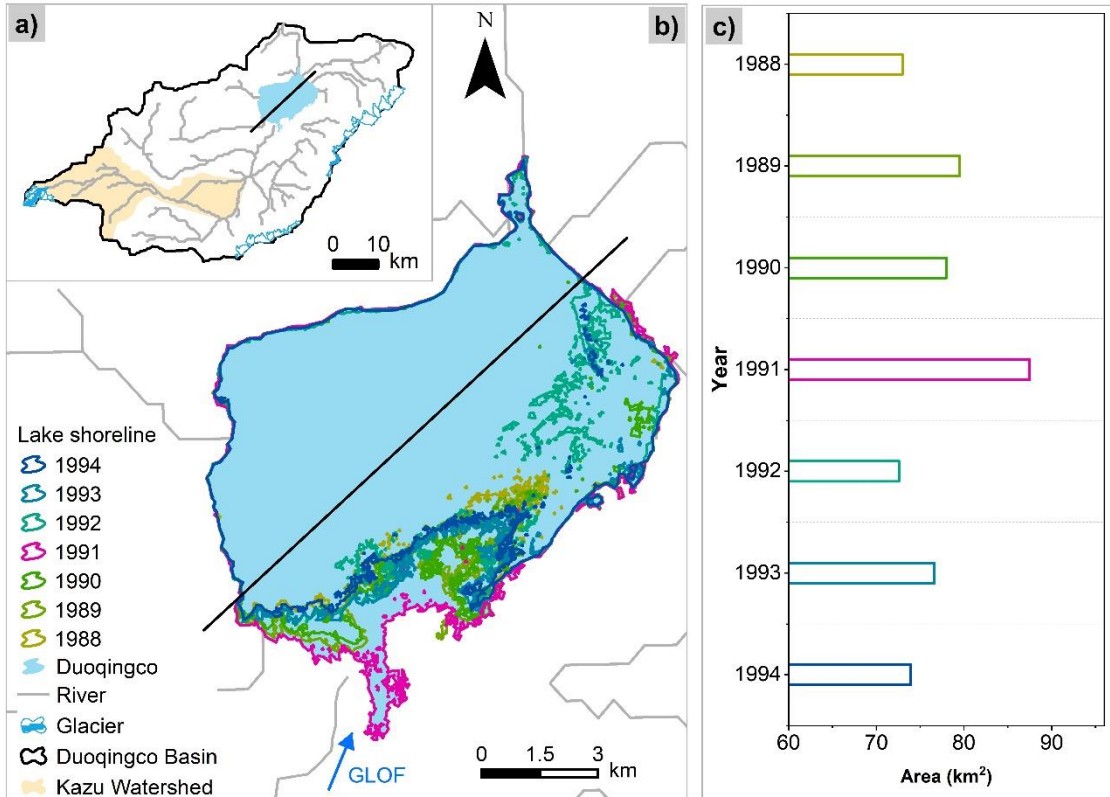

**Figure 13: Variation of the Duoqing's shoreline and water area during 1988 and 1994 interpreted from the Landsat images. a) the location of the Duoqing lake in the whole basin, b) the shoreline variation during 1988 and 1994, and c) the lake water area of 1988-1994.**

## 5 Discussion

### 5.1 Contribution of Hydrometeorological Conditions to the GLOF

Worldwide, including the Himalayas, glaciers are rapidly melting in the context of changing climatic conditions and are leading to the growth of glacial lakes (Khadka et al., 2023; Nie et al., 2017; Shugar et al., 2020). The long-term climatic data (1957-2017) from Pali meteorological station shows an increasing temperature trend of 0.017 °C per decade (**Fig. 2**). Between 1977 and 2020, following the temperature increment, glaciers in our study catchment shrunk by more than 3 km$^2$. In broader terms, when we look at the glacier mass balance rate of our study region, the mass loss rate increased to -0.52 meters of water equivalent (m.w.e.) per year from -0.35 m.w.e / year (Maurer et al., 2019; Nie et al., 2020). Similarly, the long-term precipitation from 1957 to 2017 also shows an increasing trend, contrary to the long-term tendency along the HKH, as illustrated by Ren et al. (2017). Although, after 1981, the precipitation records in the current study area show a slightly decreasing trend. When the annual climatic data for 1991 was compared with the historical mean (five-year moving average),





the precipitation and temperature were in almost the same range as the historical records (**Fig. 14a**). However, the month-wise analysis of 1991's meteorological condition with the long-term monthly average (1957-2017)- referred to as 1991 temperature and precipitation anomaly here- sheds light on some interesting fluctuations (**Fig. 14a and 14b**). The 1991 temperature anomaly clearly shows that the monthly temperature exceeded the long-term mean since February and maintained the tendency

for the entire period except for a few months. In October 1991, the mean monthly temperature exceeded the long-term mean by greater than 0.5°C (**Fig. 14a**).

Similarly, the 1991 precipitation anomaly depicts a clear underlying monthly precipitation trend. Despite the cumulative precipitation in October being more than 20 mm below the extended mean value, the data of the preceding months show a significant increment of 40 and 50 mm in August and September, cumulating about 100 mm more than in the average period

(**Fig. 14b**). From our analysis, we postulate that climate warming and rise in precipitation contributed to glacier retreat and instability, resulting in GLOF. On one hand, climate warming melts the glacier ice, causing the expansion of glacial lakes and increasing hydrostatic pressure over the moraine dam. On the other hand, climate change (both increase in temperature and precipitation) is regarded to enhance the mass wasting processes and increase the occurrence of snow-avalanches/glacier calving, which have been recorded in the Himalayas (Ballesteros-Cánovas et al., 2018). This is proposed as one of the possible

scenarios for the occurrence of Rijieco GLOF. However, no such significant anomalies were observed in the daily temperature and precipitation records, both in-situ station and satellite data.

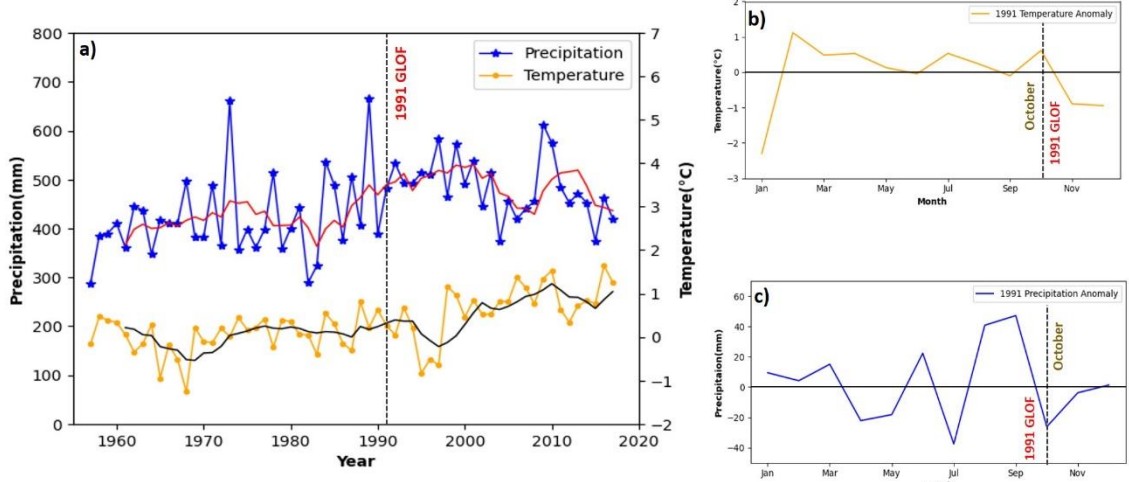

**Figure 14: Annual and monthly anomalies of precipitation and temperature from the Pali meteorological station. a) Interannual variation (1957-2017) of temperature and precipitation with the historical mean (five-year moving average) indicated by black and**
**red lines, consequently. b) The temperature anomaly of 1991 compared with long-term average temperature. c) The precipitation anomaly of 1991 compared with long-term average precipitation. The long-term average indicates the time from 1957 to 2017. The dotted vertical lines indicate the occurrence of GLOF in 1991, whereas the horizontal lines in b) and c) represent zero lines.**



## 5.2 Considerations in hydrodynamic simulation of GLOF

Our simulation is based on Newtonian flow assumptions; however, in most cases, GLOFs turn into hyper-concentrated or
debris-laden floods (Sattar et al., 2022), depending on watershed characteristics like stream gradient and availability of loose
sediments for entrainment (Cui et al., 2010; Westoby et al., 2014b). Modeling the non-Newtonian fluids requires more
information on flood rheology (Worni et al., 2014), which is beyond the scope of this study. Instead, we used multiple bulking
factors for the clear water hydrograph, varying them between 20% and 40% (Gusman, 2011; Jong-Levinger et al., 2022) to
account for sediment incorporation in the flow. We found the bulking factor of 1.3 more suitable when compared with the
inundation extent in Landsat satellite imagery just after the flood and with the historic flow traces during the field assessment.
Similarly, the peak discharge obtained from our dam break modeling resulted in 2900 $m^3$/s, identical to the maximum outflow
calculated from Froehlich (2008) relation. We used empirical relationships to calculate breach width and formation time using
Froehlich (1995) equations. The calculated top breach width of the moraine and the one measured in the field were within the
proximity of 3 m, thus warranting the use of Froehlich's equation in our study. We did not use a physical-based dam erosion
model because that would require detailed material properties and geophysical investigation of material properties that remain
unknown in this study (Costa & Schuster, 1987; Gouli et al., 2023).

We used open-access DEMs in this study – 8 m HMA DEM and 30 m NASADEM- to carry out the outburst simulation.
Although the HMA DEM is very high-resolution compared to other terrain models, there are plenty of voids in it (Sattar et al.,
2022), which makes it unsuitable to conduct hydrodynamic simulation over a large area. Consequently, the downstream
propagation of the flood was carried out in the 30 m NASADEM, and the simulation results look reasonable. The simulation
results could have been enhanced if a very high-resolution DEM - such as those prepared from UAV - had been used in the
narrow channels and the alluvial fan. Wang et al. (2024) also highlighted the importance of such high-resolution UAV-
generated DEM for better representation of the flood simulation in the narrow river channels of the Poiqu River basin, although
we did not face a flow discontinuity problem in our case with free DEMs too. Similarly, manning's roughness is one of the
critical parameters in flood modeling, so we carried out sensitivity analysis using a range of roughness coefficients '0.07', '0.1',
and '0.15', primarily by analyzing the surface types through satellite imagery and values given by Chow (1959). The minimum
depth (< 8 m) and maximum velocity (>10 m/s) were obtained in a '0.07' roughness value with a minimum travel time of 4
hours. Likewise, for the '0.1' roughness value, the average depth and velocity were obtained at <10 m and >8 m/s.
Notwithstanding, based on satellite imagery and approaches taken by previous modeling (Wang et al., 2018), we found a 0.15
roughness value more appropriate in our case.

## 5.3 Future Hazard of Rijieco and GLOF Exposure

The future GLOF from Rijieco glacial lake will most likely be of lesser magnitude. The lake had already attained its maximum
level in 1991, just before the GLOF occurred, and it is currently already detached from the parent glacier (**Fig. 3**). Crevassed
and hanging ice mass can be seen laden over the massifs (**Fig. 3a**); however, it is less likely they will hit the lake because of





the low reach angle (Allen et al., 2019) - except the same slope from which we anticipate that an avalanche had trigger - and comparatively, their less volumetric scale when matched to the lake. Similarly, the slope of the parent glacier is also not very steep, nor does it have significant fetch (Sakai et al., 2009), so it is less likely that glacier calving will result in the re-failure of the lake. Moreover, the moraine dam of the lake has already breached by 75×31 m2; the lake can easily adjust additional discharge until the overtopping wave/spilled water further incises the barrier, exceeding the shear strength of its material

(Worni et al., 2014). Despite that, a few large fractures and cracks are present on the permafrost slope (**Fig. 8e, 8f, and 8g**), which can be sources of future GLOF triggers or catastrophic cryospheric events like that occurred in Chamoli in February 2021 (Shugar et al., 2021). To properly assess the risk of future hazards, it is necessary to carry out more extensive field investigations through which lake bathymetry, moraine dam material properties, and mass flow status should be quantified. Furthermore, there is a lack of in-situ stations in the lake's watershed area to measure the hydrometeorological data and glacier

velocity and monitor lake water level. Installing such field-based instruments would be essential to monitor future hazards and analyze the underlying factors that increase GLOF susceptibility.

## 6 Conclusions

Using a comprehensive methodology of remote sensing interpretation, field survey, and hydrodynamic modeling, we performed a site-specific investigation of a historical GLOF event that happened at the Rijieco moraine-dammed lake in the

eastern Himalayas. The event's timing causes, and ecological impacts are examined with multiple data sources, including decades of remote sensing imagery, high-resolution DEMs, on-site measurement data, and locally recorded meteorological data. The detailed 2D dam break and hydrodynamic modeling with sensitivity analysis are used to reconstruct the outburst flood's cross-sectional hydrographs and downstream propagation process. The main findings of this study are listed as follows:
a. From 1977 to 1991, the lake area constantly increased from 0.34 km$^2$ to 0.43 km$^2$ (by image analysis) or 0.46 km$^2$ (by field),

strongly correlated with the decreasing glacial area. At the end of 1991, a reduction of 44% in the lake area was observed, and then the area just had a minor change. Field witness of a dam breach of 31 m (height) × 75 m (top width) confirms a historical GLOF event. The first occurrence of traces of fresh water on the satellite image on October 7, 1991, indicates the GLOF event happened most likely in the first week of October in that year. During the GLOF, the lake area shrank by 0.2 km$^2$, and nearly 6 million m$^3$ of water was released. Using the HEC-RAS, the peak discharge at the dam site was computed at 2900 m$^3$/s, and

the maximum inundation depth and velocity reached 11 m and 8 m/s, respectively. When the flood traveled 30 km into the downstream Duoqing lake, the resulting peak discharge decreased first to around 200 m$^3$/s at the fan's apex and then 45 m$^3$/s at the fan's apron.
b. The outburst flood imposed short-term geomorphic and long-term ecological impacts on a Tibetan inland lake system. The flood entrained and transported huge amounts of water and sediments along the channel and into the Duoqing lake – covering

the entire vegetation along some flow sections with debris and increasing the lake area by 8.5 km$^2$ immediately after the 1991 GLOF. The GLOF discharge caused the abnormal expansion. The water area of the Duoqing lake was reduced by 17% just in

the year before the event (1992). The sediments cover an area of $1.0\times10^6$ m$^2$ of the channel and flood plain from the dam to the middle stream, where the vegetation has not recovered even 32 years after the GLOF. Similarly, a large portion of the alluvial fan and the Duoqing's green lands has been silted by the GLOF's sediment.

c. Since the GLOF happened in the early 1990s, the actual factor that triggered it was unknown. We propose three scenarios in sequential order of their probability: a) an ice avalanche from nearby hillslopes when the glacial lake held water at its maximum capacity, b) glacier calving, and c) piping/ seepage that could have caused a dam breach. The recorded precipitation and temperature data at the local meteorological station show a positive precipitation anomaly (+100 mm compared to the long-term mean) in the preceding two months of the GLOF. In contrast, a positive temperature anomaly occurred in the month
of GLOF. Such anomalies indicate possibilities of glacier instabilities due to a combination of excess precipitation and rapid melting water.

   d. As a specific moraine-dammed lake on the Tibetan Plateau, Rijieco can be an ideal site for studying the long-term interaction between climate change and the environment. In-situ monitoring equipment and high-resolution remote sensing data are required to observe hydrometeorological conditions, glacier change, and landscape evolution. The findings should be utilized
to make future hazard assessments more robust, create awareness, and enhance the capacity of authorities and downstream communities to make them resilient.

**Acknowledgements**

This research was funded by the Key Research and Development Program of Tibet Autonomous Region (XZ202301ZY0039G), the Second Tibetan Plateau Scientific Expedition and Research Program (2019QZKK0902), and the Science and Technology
Research Program of the Institute of Mountain Hazards and Environment, Chinese Academy of Sciences (IMHE-ZDRW-01). MRG acknowledges the 'ANSO Scholarship for Young Talents' for his postgraduate study.

**Author contribution**

KHH contributed to the conceptualization, supervision, funding acquisition, formal analysis, and writing. MRG contributed to methodology, image interpretation, visualization, the HEC-RAS simulation, and writing. HL and YN contributed to the
investigation of the Rijieco and Duoqing lakes. KHH, HL, SL, PL, and XPZ contributed to the field survey. HL, MRG, and YS contributed to satellite and background data collection. All authors contributed to the preparation and editing of the paper.

**Code/Data availability**

The meteorological data were obtained from the China Meteorological Administration. All other datasets and models used in this study are open-source and freely available from the sources cited in the text.



**Competing interests**

The contact author has declared that none of the authors has any competing interests.

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
