# Peer review of "Timing, Causes, and Ecological Impacts of the 1991 Glacial Lake Outburst Flood at Rijieco in the Eastern Himalayas"

_EGUsphere, 2024_

## Referee Comment (RC1)

**Paper: egusphere-2024-884, Timing, Causes, and Ecological Impacts of the 1991 Glacial Lake Outburst Flood at Rijieco in the Eastern Himalayas.**

**General Comments**

Originality: Fair

This study seeks to investigate a historical Glacial Lake Outburst Flood (GLOF) that occurred in 1991 at lake Rijieco. They aim to combine remotely sensed data with climatic data and hydrological modelling to determine the magnitude of the GLOF along with its potential triggers. There is merit in the historical investigation of GLOF events to further our understanding of landform susceptibility and trigger mechanisms. The modelling of the 1991 GLOF magnitude and impact, although not novel does provide a useful insight into reconstructing this event, however this is overshadowed by the papers poor scientific quality in other aspects.

Scientific Quality: Poor

Scientific understanding:

I note the authors have sought to address knowledge gaps in the current understanding of GLOF's however they have omitted multiple key references explaining these and have drastically oversimplified key characteristics associated with GLOF's. Critically, there is no key distinction found herein between "susceptibility" and "triggers", hindering the scientific argument underpinning this paper. Susceptibility is linked to the glacial lake dam and can relate to: lake volume, dam composition, dam morphology and lake freeboard. This paper fails to address these elements of the dam at Rijieco and this is critical when assessing any glacial lake related hazard. Trigger mechanisms refer to an event or series of events that instigate a hazard, which in the case of GLOFs comprise: avalanches, calving events, melting of ice cores, seepage, inappropriate engineering works and earthquakes (debateable). Although the authors try address these they are far too definitive in their interpretations without sufficient evidence of the triggers. Given that the event occurred over 30 years ago, a rigorous method should be applied to consider which if any of these triggers is more likely to occur than the other. Of note is that the explanation of the hazard chain is not in fact feasible (Figure 7), displaying a rockfall of 1 vertical kilometre and 8 horizontal kilometres into the lake. This figure and associated text in particular oversimplifies the hazard chain with little evidence to support these inferences from current datasets and omits extensive work done by multiple academics to explain these hazards in detail.

Furthermore, process identification forms a key component of any hazard assessment as without a proper understanding of the susceptibility of a land system as well as potential triggers for an associated hazard at a site-specific scale, the assessment is unlikely to be representative (Emmer et al., 2013; Racoviteanu et al., 2022). A glacial hazard assessment should aim to combine a suite of datasets, both remotely sensed and (if warranted) geophysical to accurately ascertain the factors inherent to the risk (Richardson and Reynolds, 2000; Reynolds, 2006; Racoviteanu et al., 2022). This should aim to link the susceptibility of landform to the potential trigger mechanisms of that site with consideration of how those factors may develop with climatic changes over a number of timescales (Wang et al., 2018; Emmer et al., 2020; Racoviteanu et al., 2022; Reynolds, 2023). Given the justification for this

study is that of hazard assessment, I would expect to see a  comprehensive hazard assessment including the following steps (Reynolds, 2023):

1. Desk study
2. Gap analysis
3. Event mapping
4. Process identification
5. Neo-tectonic provinces and seismicity
6. Regional geological analysis
7. Landslide susceptibility mapping
8. Glacial hazard assessment
9. Hydrological analyses
10. Sediment management
11. Other analyses identified from gap analysis

Although elements of this are addressed I would argue that they are not done in a sufficient way to truly represent a hazard assessment.

Ultimately this study does not undertake a rigorous hazard assessment for this site and based upon inconsistencies in their explanation of GLOF mechanics. There are elements of GLOF reconstruction but these are undermined by the authors explanations of likely GLOF mechanisms. I would argue that this work does not truly assess either the historical GLOF or any other hazards in the region.

Data collection and analysis:

Although interesting, the remotely sensed data fails to acknowledge uncertainties, particularly in the resolution of the various sensors used. Furthermore, there are instances where higher resolution sensors were freely available from USGS and were not used (Sentinel, etc.,), this hinders interpretation of processes occurring. I feel there is a lack of acknowledgement in regard to the resolution of sensors and the scale of the changes to the geomorphology and development of these landforms, this is critical for a rigorous method. This has led to a large number of assumptions as to the nature of the GLOF throughout the paper based upon historical imagery that is not of a sufficient resolution, hindering the argument.

The field campaign (Figure 8) does not in fact yield useful data in terms of hazard assessment and to accurately assess any hazard a comprehensive assessment should be undertaken following established guidelines (see Reynolds, 2023).

Significance: Poor

This manuscript does not contribute to changing our scientific understanding of a subject substantially or to introducing new practical applications of broad relevance. I agree that this site warrants investigation, to a degree, but the application of this has note been done sufficiently to draw any meaningful conclusions.

Presentation Quality: Poor

The figures presented within this manuscript have numerous inconsistencies and are not of publication standard. This is explained in detail in the specific comments.

**Specific comments**

32: Consider reframing global warming to "climate change"

38: Earthquakes as a trigger mechanisms of a GLOF is debatable and hasn't been directly observed (see Wood et al., 2024)

41 What is the timeframe for these reported events? This gives an indication of frequency.

46 Again, this requires a timeframe of these records to give an indication of frequency.

54 I feel this oversimplifies the hazard chain occurring here and could warrant further explanation given the importance of this case.

60/61 These sentences could be combined as they effectively say the same thing.

63 Why is this the case? Further details are required to back up this statement.

65 I would also draw on other studies from HMA and potentially south America (Racoviteanu et al., 2022; Richardson and Reynolds; 2000; Reynolds, 2006; 2017; 2022; 2023; Wang, 2020)

66 Again key studies are missing (Westoby et al., 2014; 2015; Nei et al., 2020; Majeed et al., 2021)

69 Yes, from a remote sensing perspective but what about geomorphological evidence (Westoby et al., 2014) this requires further clarification.

97 What are these glacier's names, shape, area etc?

Figure 1:

- Given that you are discussing the glacier, lake and alluvial fan I find this figure not especially useful.
- I would suggest having the overview of the region as a smaller insert with greater attention being given to the glacier and alluvial fan.
- This would benefit from geomorphological mapping clearly showing the components of the glacier system as they are all integral in discussing the nature of any GLOF (Racovitnaeu et al, 2022).
- The current insert b does not clearly show the current glacier dam, etc well so it is hard to visualise the study site.
- I would suggest using the Maxar <1m imagery available in ArcPro as the basemap if possible as the resolution is likely to be better.

122-125 This is not data sources and is more methodology, suggest moving to a different section.

Table 1

- I would like to see the resolution of your data sources included here as this will directly influence the uncertainty associated with your interpretations and will have an impact on the equations used herein.
- Would also like to see additional metadata of the imagery and why they were selected.

135 The resolution of Landsat varies between missions so that will change the resolution of any interpretations you make from this. This must be quantified and considered to draw any meaningful conclusions.

135 Could you have used sentinel data?

144 I would suggest cross checking between multiple experts given the manual digitisation, one expert is not sufficient.

192 How do you know these advances are recent?

Figure 3

- The colours used for text make this very hard to read, consider bounding boxes around text
- Panel A adds very little instead a detailed geomorphological map would be of benefit here
- What evidence do you have of previous lake level?
- Pannels e and f do not add to this figure

250 You cannot say retreated significantly without using metrics to quantify this. How fast and where did it retreat, you should have this data from your analysis.

Figure 4

- Red watershed colour cannot be seen on false colour images, please change.
- All colours are hard to distinguish.
- I would prefer to see 2 figures here, one showing the changes to glacier size overlayed year on year for all data and one for lake size evolution for all years overlayed. This would be far clearer as the current extents are very hard to interpret from this figure.

262 I would be wary of assigning a date given the temporal gaps in your data, better to leave as a range of dates where the flood could have occurred.

Figure 5

- Similar comments to the previous figures at this extent and resolution the data is very hard to interpret.
- The alluvial fan should be zoomed in on and the polygon delineating its location should be around its margins not just its general area.
- The symbology for North etc, is inconsistent across the figures.

272 What field measurement?

273 Saying that the lake was partially breached is an inference not a direct observation from the event. You do not know whether this occurred in this exact manor so careful consideration needs to be taken in definitions and available evidence.

285 By definition a glacier can only retreat towards a terminus otherwise it would be in advance.

294-296 You have confused your triggers for a GLOF with the mechanisms to which a GLOF occurs. Your scenario A would cause a displacement wave resulting in either a displacement

flood or a seiche burst flood, the trigger is the avalanche where as the displacement wave is the mechanism of flood. The same applies to scenario B where the calving (the trigger) causes a displacement wave (the mechanism). This section needs to be rewritten to carefully define these processes. See: Racovitnaeu et al, 2022; Richardson and Reynolds, 2000; Westoby 2014; etc.,

297 This is current data not historical so is a very large assumption and not definitive.

302 This a very large assumption and is dependant on the trigger mechanism. Furthermore, there are multiple ways the flood can propagate so this is tenuous at best.

306 Again this is current evidence not historical so is a bit of a reach.

315 This pool could have formed from other methods so piping cannot be directly assumed from your data.

Figure 7 This figure is not of publication standard the issues are as follows:

- The figure caption does not match the content displayed in the figure.
- The large main schematic has vertical access values that are unrealistic. It is overly simplified and arguably doesn't add much to any explanation or overarching story and is of poor quality.
- The remotely sensed images are off-nadir.
- The scenarios are a oversimplification of the processes occurring and do not go into detail as to how these triggers impact the nature of the GLOF. See Westoby, 2014.
- Text should not be coloured.
- There are numerous blank spaces within the figure that could be removed.

Figure 8

- Overall although a nice series of photographs this does not work as an effective assessment, a number of these could be used to give context to the site, however this does not add to the paper as a standalone figure.
- Caption: the surrounding features do not make it susceptible to a GLOF, these are trigger mechanisms, the dam that burst is the landform that is susceptible.
- Panel a, the insert map has no scale and I do not know what the colours mean.
- Panel b: This is wildly speculative as there is no direct evidence for this.
- The references to ice and rock cracks are again spurious and do not necessarily indicate a hazard purely from photographic evidence. To make this kind of inference geotechnical surveying should be undertaken as a minimum.

Figure 9

- This again is a large leap considering the resolution of the data and cannot be used as a definitive inference of processes occurring.

400 – 404 This section drastically oversimplifies the controls on lake formation, development of glaciers in the regions, the susceptibility of the landform in the event of a GLOF and the role of trigger mechanisms. This also fails to address key literature.

442 – 450 The reduction in any future GLOF risk is obvious given the lake has drained and not refilled and will likely self-regulate due to the dam breach.

450 Again, this is a drastic oversimplification of events and comparing an observed GLOF with potential future triggers at a site that has already had a GLOF does not provide justification.

452 Yes, agreed, more does need to be done to monitor these landforms but why this one, given it has already drained? I would argue that the hazard presented in this study has not been sufficiently assessed and further care should be taken to define these hazards with comprehensive assessments as outlined by Racoviteanu et al., 2022; Richardson and Reynolds; 2000; Reynolds, 2006; 2017; 2022; 2023.

460 You cannot say from your data the cause of the GLOF. You can describe different trigger mechanisms resulting in different styles of LGOF but you do not have sufficient evidence in the data shown to accurately make the interpretations you have.

480 Yes, you do propose 3 possibilities, however, your explanation of these phenomenon along with your methods for inferring which was most likely are insufficient to make any conclusions.

487 I strongly disagree with this statement. A GLOF has already occurred here, thus time and money is better spent on glacial lakes that have not drained and still could present a hazard. There is argument for historical reconstruction of GLOF's however I think you are overselling the significance and novelty of this site.

489 Your findings and methods are not sufficient for a hazard assessment, they are not robust and do not follow well documented doctrine. Furthermore, they oversimplify the various components of a GLOF and make large leaps of assumption without sufficient evidence.

**References:**

Emmer, A. and Cochachin, A., 2013. The causes and mechanisms of moraine-dammed lake failures in the Cordillera Blanca, North American Cordillera, and Himalayas. AUC GEOGRAPHICA, 48(2), pp.5-15.

Emmer, A., Harrison, S., Mergili, M., Allen, S., Frey, H. and Huggel, C., 2020. 70 years of lake evolution and glacial lake outburst floods in the Cordillera Blanca (Peru) and implications for the future. Geomorphology, 365, p.107178.

Majeed, U., Rashid, I., Sattar, A., Allen, S., Stoffel, M., Nüsser, M. and Schmidt, S., 2021. Recession of Gya Glacier and the 2014 glacial lake outburst flood in the Trans-Himalayan region of Ladakh, India. Science of the Total Environment, 756, p.144008.

Nie, Y., Liu, W., Liu, Q., Hu, X. and Westoby, M.J., 2020. Reconstructing the Chongbaxia Tsho glacial lake outburst flood in the Eastern Himalaya: Evolution, process and impacts. Geomorphology, 370, p.107393.

Racoviteanu, A.E., Nicholson, L., Glasser, N.F., Miles, E., Harrison, S. and Reynolds, J.M., 2022. Debris-covered glacier systems and associated glacial lake outburst flood hazards: challenges and prospects. Journal of the Geological Society, 179(3), pp.jgs2021-084.

Reynolds, J.M., 2006. Role of geophysics in glacial hazard assessment. First Break, 24(8).

Reynolds, J.M., 2017. Integrated Geohazard Assessment as part of Climate Change Resilience and Disaster Risk Management in the hydropower sector in high mountain environments. In HYDRO 2017 Conference Proceedings, Seville, Spain, 9-11th October 2017.

Reynolds, J.M., 2022. Integrated geohazard assessments to increase the resilience of hydropower schemes to future natural disasters in the Himalayan Region. In Proceedings.

Reynolds, J.M. and Reynolds Geo-Solutions Ltd, U.K., 2023. The role of integrated geohazard assessments in disaster risk management. International Journal of Hydropower & Dams, 30(1), pp.43-56.

Richardson, S.D. and Reynolds, J.M., 2000. An overview of glacial hazards in the Himalayas. Quaternary International, 65, pp.31-47.

Richardson, S.D. and Reynolds, J.M., 2000. Degradation of ice-cored moraine dams: implications for hazard development. IAHS PUBLICATION, pp.187-198.

Wang, W., Gao, Y., Anacona, P.I., Lei, Y., Xiang, Y., Zhang, G., Li, S. and Lu, A., 2018. Integrated hazard assessment of Cirenmaco glacial lake in Zhangzangbo valley, Central Himalayas. Geomorphology, 306, pp.292-305.

Wang, S., Che, Y. and Xinggang, M., 2020. Integrated risk assessment of glacier lake outburst flood (GLOF) disaster over the Qinghai–Tibetan Plateau (QTP). Landslides, 17, pp.2849-2863.

Westoby, M.J., Glasser, N.F., Brasington, J., Hambrey, M.J., Quincey, D.J. and Reynolds, J.M., 2014. Modelling outburst floods from moraine-dammed glacial lakes. Earth-Science Reviews, 134, pp.137-159.

Westoby, M.J., Glasser, N.F., Hambrey, M.J., Brasington, J., Reynolds, J.M. and Hassan, M.A., 2014. Reconstructing historic Glacial Lake Outburst Floods through numerical modelling and geomorphological assessment: Extreme events in the Himalaya. Earth Surface Processes and Landforms, 39(12), pp.1675-1692.

Westoby, M.J., Brasington, J., Glasser, N.F., Hambrey, M.J., Reynolds, J.M., Hassan, M.A. and Lowe, A., 2015. Numerical modelling of glacial lake outburst floods using physically based dam-breach models. Earth Surface Dynamics, 3(1), pp.171-199.

Wood, J.L., Harrison, S., Wilson, R., Emmer, A., Kargel, J.S., Cook, S.J., Glasser, N.F., Reynolds, J.M., Shugar, D.H. and Yarleque, C., 2024. Shaking up assumptions: Earthquakes have rarely triggered Andean glacier lake outburst floods. Geophysical Research Letters, 51(7), p.e2023GL105578.